# Exploring Knowledge and Perceptions of Polio Disease and Its Immunization in Polio High-Risk Areas of Pakistan

**DOI:** 10.3390/vaccines11071206

**Published:** 2023-07-05

**Authors:** Muhammad Atif Habib, Farhana Tabassum, Imtiaz Hussain, Tooba Jawed Khan, Nazia Syed, Fariha Shaheen, Sajid Bashir Soofi, Zulfiqar A. Bhutta

**Affiliations:** 1Centre of Excellence in Women and Child Health, Aga Khan University, Karachi 74800, Pakistansajid.soofi@aku.edu (S.B.S.); 2Institute for Global Health and Development, Aga Khan University, Karachi 74800, Pakistan; 3Centre for Global Child Health, The Hospital for Sick Children, Toronto, ON M5G 0A4, Canada

**Keywords:** polio, polio immunization, knowledge and perceptions, Pakistan

## Abstract

Pakistan is one of the few countries where poliovirus transmission still persists, despite intensive efforts to eradicate the disease. Adequate vaccination coverage is essential to achieve polio eradication, but misconceptions about polio vaccines have hindered vaccination efforts. To address this issue, we conducted a mixed-methods study to explore knowledge and perceptions regarding polio disease and immunization in high-risk areas of Pakistan. We collected quantitative data from 3780, 1258, and 2100 households in Karachi, Bajaur, and Pishin, respectively, and supplemented this with qualitative data from focus group discussions and in-depth interviews. Our findings reveal a high level of awareness about polio and its immunization; however, misperceptions about the polio vaccine persist, leading to refusal for both polio vaccines and routine immunizations. Our study provides up-to-date data on knowledge and perceptions of polio and its immunization and identifies critical gaps. These findings can inform the development of future strategies and innovative approaches to improve the success of the polio program in Pakistan.

## 1. Introduction

The global polio eradication initiative (GPEI), launched in 1988, has successfully eradicated and contained all wild, vaccine-related, and Sabin polioviruses globally, except in Pakistan and Afghanistan [1]. The GPEI was established in 1994 in Pakistan and has substantially reduced the country’s polio burden [2]. However, despite concerted national efforts, the transmission of wild poliovirus continues to occur, and Pakistan continues to be affected by wild poliovirus type 1 (WPV1) and circulating vaccine-derived poliovirus type 2 (cVDPV2) [3]. The lowest number of cases reported was 8 in 2017, but a rise in polio cases was observed in the following years, with 12 cases in 2018, 147 cases in 2019, and 84 cases in 2020. In 2020, the polio surveillance system in Pakistan identified poliovirus circulation in 38 districts of Pakistan, including the high-risk areas of Karachi, Northern Sindh, Southern Punjab, Peshawar, and the Quetta block. In 2022 virus circulation is ongoing, and to date, Pakistan has 20 positive cases, all confined to the hard-to-reach Waziristan area [2].

In Pakistan, the GPEI has implemented numerous vaccination campaigns annually to contain polio transmission. About 250,000 frontline health workers go door-to-door to ensure every child under five is vaccinated against polio and receives the oral polio vaccine (OPV) [4]. In addition to the OPV given during the campaigns, four doses of the OPV are also offered at birth and the ages of 4, 10, and 14 weeks as a part of routine immunization in the National Expanded Program of Immunization (EPI) [5]. In 2015, the Government of Pakistan also included one dose of Injectable Polio Vaccine (IPV) to be given at 14 weeks in the national EPI to further boost immunity among children [6]. To eradicate Polio from Pakistan, all children must receive a sufficient number of polio doses during the campaigns and routine immunizations [7]. Failure to do so will result in continued poliovirus circulation in Pakistan [8].

Despite numerous awareness campaigns and initiatives through local and international stakeholders and partners, fears and misperceptions about polio vaccines persist among high-risk communities [9,10]. The literature suggests that misperceptions based on cultural and religious beliefs and customs are the critical barrier to polio vaccination [10,11,12]. Misconceptions surrounding the oral polio vaccine, propagated by religious leaders, extremist groups, and organizations in Pakistan, have emerged as significant barriers to the uptake of the vaccine [12,13,14]. Another study proposed that the pervasive use of social media has led to the growing prevalence of religious fatalistic beliefs coupled with the dissemination of misinformation, disinformation, and fake news, presenting a significant risk to the adoption of protective behaviors [15]. Prevalent rumors in high-risk communities include claims that the OPV is anti-Islamic, contains impermissible haram ingredients, causes infertility, and that the Global Polio Eradication Initiative (GPEI) is a foreign agenda [11,12,13,14].

Robust communication strategies are required to counter these misconceptions and enhance the knowledge of parents and caregivers about polio and its vaccines. However, current data on insights and perceptions about polio and its immunization in the high-risk population are lacking [11]. Therefore, we conducted a mixed-method study in high-risk areas for polio in Pakistan to identify the current knowledge and perceptions about polio and its vaccination. In addition, this study estimates the current coverage of routine immunization and uptake of the oral polio vaccine and compares the existing knowledge, perceptions, and vaccine coverage with previous findings [11].

## 2. Methodology

We used a mixed-methods approach to achieve the objectives of the study. Quantitative data were collected through a cross-sectional survey conducted at the household level, while qualitative data were obtained through focus group discussions (FGDs) and in-depth interviews (IDIs) with the targeted groups. The study took place from January 2020 to March 2020 in high-risk areas for polio in Pakistan. Specifically, we collected data from Karachi in Sindh province, Pishin district in Baluchistan province, and Bajaur district in Khyber Pakhtunkhwa (KP) province in Pakistan (Figure 1).

We employed a multi-stage sampling technique to draw a random sample from each area (i.e., Karachi, Pishin, and Bajaur) and considered the union council (UC) as our cluster. A UC is the smallest administrative unit in Pakistan, which usually has a population of 20,000 to 30,000. In the first stage, we selected 18 UCs from Karachi, 10 from Pishin district, and 6 from Bajaur district (Table 1). After choosing the UCs, we utilized the WHO 30X7 technique [16] to achieve the sample size within each union council for the quantitative part of the study. This approach provided us with a sample size of 210 in each union council, which is a suitable sample to assess the coverage of vaccines and the knowledge of parents and care providers [17]. The overall sample sizes for Karachi, Pishin, and Bajaur were 3780, 2100, and 1260, respectively. In order to determine the 30 areas and 7 households for our study, we relied on the micro plans utilized by the polio eradication program. These micro plans are comprehensive plans created for each union council outlining the specific areas where vaccinations are administered and identifying households with children under 5 years of age. From the provided lists, we randomly selected 30 areas and from each of the selected areas, 7 households were selected with children under 5 years of age for the interview.

To collect the necessary data, we designed a structured questionnaire specifically for the household survey. The questionnaire was carefully translated into Urdu (the national language) and then back-translated into English to ensure accuracy and consistency. Prior to the actual survey, we conducted a thorough pre-test of the questionnaire in areas that were not part of the survey sample. Based on the feedback and insights gained during the pre-test, we made necessary updates and refinements to the questionnaire. For the data collection process, we recruited local teams of experienced female data collectors who were well-versed in the local culture, language, and had prior data collection experience. To equip them with the required skills and knowledge, we conducted a comprehensive 4-day centralized training session on survey methodology, sampling techniques, and proper administration of the survey. Following the training, the data collectors underwent 2 days of mock field interviews to further enhance their understanding and proficiency. Once fully prepared, the survey teams commenced data collection at the household level. Before every interview, written consent was obtained from the respondents, which included mothers, fathers, or caregivers. The pen-and-paper personal interview (PAPI) method was used for data collection.

In addition to the quantitative component, we conducted qualitative research through focus group discussions (FGDs) and in-depth interviews (IDIs) in the locations of Karachi, Pishin, and Bajaur. The qualitative component involved engaging various participants, including decision-makers at the household level (such as fathers, mothers, and mothers-in-law), influential members of the community, and healthcare providers such as community health workers and vaccinators (Table 2). The selection of participants for the IDIs and FGDs was performed through purposive convenience sampling.

Earlier, the qualitative data collection tools were developed and pre-tested specifically for the qualitative component, including FGD and IDI guidelines. Through a series of pre-tests involving different target groups, we ensured the content was clear, relevant, and appropriate for the research objectives. Skilled female moderators and note-takers, along with local facilitators, were responsible for collecting the qualitative data, which ensured a comprehensive and well-rounded approach to data collection.

## 3. Data Analysis

The data were analyzed using STATA version 16 software. A descriptive analysis was carried out to assess the knowledge, attitudes, and practices regarding the polio vaccine, reasons for refusal, and current immunization status. Due to the categorical nature of the observed variables, the data were summarized as frequency and percentages. In addition, a comparative analysis was performed separately for the three districts to assess changes in key indicators over the two survey rounds, i.e., 2012 and 2020. A comparison between KAP data obtained from subjects was performed using the Chi-square test. A value of *p* < 0.05 was considered significant.

For the qualitative data, we used topical and content analysis to convert the responses into emerging and meaningful categories for each target group using a deductive approach. All transcriptions were comprehensively reviewed and coded. NVIVO software (version 12) was used to analyze the qualitative data. We used these qualitative data findings to complement the quantitative part of the study and to strengthen the overall findings of this study.

## 4. Results

### 4.1. Quantitative Component

In the quantitative component of the survey, the data was collected from 1258, 2100, and 3780 households from Bajaur, Pishin, and Karachi, respectively. A total of 8025 households were surveyed in high-risk areas for polio. Table 3 summarizes the demographic information from the target areas. The average household size was 9.1 people for Bajaur, 12.5 for Pishin, and 7.8 for Karachi. The highest proportion of illiterate people was found in Pishin (84.2%) and the lowest was in Karachi (44.1%).

The findings reflected that most of the respondents had heard about polio in all three site areas: 99% in Karachi, 98% in Bajaur, and 97% in Pishin. However, their knowledge about polio transmission was poor. The primary information sources about polio were reported to be television in Karachi (47%) and Pishin (39%), along with radio in Bajaur, and LHWs in Pishin (39%). Polio was considered to be a health problem for 60% to 93% of respondents. However, half of the respondents in all areas considered immunization to be a preventive method (Table 4).

This survey also tried to explore the knowledge about different causes of polio disease. The findings indicated that over a third of all people (36% in Bajaur, 40% in Pishin) referred to it as Allah’s will. Interestingly, many residential respondents in Karachi believed that the evil eye could cause polio disease. Moreover, the transmission of polio disease was linked by respondents to drinking dirty water (33%) and contaminated air (41–47%) in Bajaur and Pishin. Surprisingly, despite being relatively knowledgeable about polio, respondents in Karachi were unaware of how polio is transmitted. The data showed that the Expanded Program on Routine Immunization (EPI) was perceived as the most comprehensive preventive measure for polio (53% in Karachi, 72% in Bajaur, and 65% in Pishin).

Table 5 shows that the knowledge about the oral polio vaccine (OPV) was reportedly high, from 97% to 99%, in all three study areas. Its effectiveness for polio prevention was understood, with 69% of respondents in Pishin, 89% in Karachi, and 94% in Bajaur perceiving it as a necessary preventive measure against polio. A substantial proportion of people reported contracting polio due to not receiving the OPV. There are still concerns regarding the OPV, which was perceived as being utterly safe by 77% of respondents in Karachi and 63% in Bajaur, but only by 42% of the people in Pishin.

Reported vaccine refusal issues were also assessed. These were highest in Bajaur (42%), followed by Karachi (23%) and Pishin (20%). The most cited OPV refusal reasons indicated in the survey were concerns about the vaccine causing sterility (19% in Karachi, 68% in Pishin). This sterility misconception was compounded by other misconceptions that the vaccine was not halal or was impermissible under Islamic laws. This was found to be highest in Pishin (45%). Additionally, the vaccine was considered to be unsafe, and children received polio drops too many times. However, the proportion of refusals influenced by local/community leaders was highest in Bajaur (21%) with respondents reporting that they had been prohibited by local or community leaders.

When specifically asked to cite reasons for not giving polio drops on National Immunization Days (NID) or campaigns, local respondents reported that the children did not receive the OPV due to the absence of the polio team or visits by LHWs (Table 6).

Polio immunization coverage reflected at different time points showed the highest immunization rate at birth was in Bajaur (94%), followed by Karachi (89%) and Pishin (85%). The survey findings on immunization card availability showed that retention of immunization cards was relatively low in Pishin (45%) and Karachi (48%). However, in Bajaur, the majority of respondents had immunization cards.

### 4.2. Comparison with a Previous Similar Survey Conducted in 2012

We compared the data of key indicators (Table 7) from our survey with a comparable survey conducted in 2012 [11]. The data revealed that respondents had a good understanding of polio, particularly in terms of its implications. However, their perceptions regarding the safety of the OPV were noticeably low.

In both surveys of 2012 and 2020, a high proportion of respondents from Karachi and Bajaur considered polio to be a significant health problem. However, there was a significant decline in this perception observed in Pishin during the recent survey of 2020. Moreover, respondents demonstrated a high level of knowledge about the OPV in both surveys. Meanwhile, no significant change was observed in the perception of the safety of the OPV in Karachi and Bajaur. However, there was a notable reduction from 49.7% to 41.9% among the respondents in Pishin who believed the OPV to be completely safe. These variations in perception and understanding of the polio vaccine in Pishin compared to other areas can be attributed to cultural differences and restrictions, lack of awareness, fears, myths, doubts, misconceptions, and misperceptions held by the locals, which were also evident in the qualitative findings.

The data on OPV refusal rates indicated a significant change in both Karachi and Pishin, with the refusal rates increasing from 5% to 23.1% and 17.4% to 25.0%, respectively. Conversely, there was a significant positive change in the availability of vaccination cards across all three sites. In Karachi, the availability increased from 26.4% to 47.7%, in Bajaur it increased from 12.9% to 78.1%, and in Pishin it increased from 6.9% to 44.5%. Additionally, we observed a substantial increment in the proportion of fully immunized children in all three areas, along with a notable reduction in the proportion of unimmunized (zero doses) children.

Overall, the data suggested changes in OPV refusal rates, improved availability of vaccination cards, and notable advancements in the immunization status of children in Karachi, Bajaur, and Pishin.

The study findings provide valuable insights into the knowledge and perceptions of polio in the surveyed areas. It was observed that a high percentage of respondents were aware of polio, although their understanding of polio transmission was limited. Television and radio were the primary sources of information, and misconceptions about polio causes were prevalent. The Expanded Program on Routine Immunization (EPI) and the oral polio vaccine (OPV) were recognized as important preventive measures, although concerns about OPV safety and religious acceptability led to vaccine refusal in some cases. Comparisons with the 2012 survey showed variations in the perceptions and understanding of polio and the OPV over time and across regions. However, immunization coverage varied in all areas, with Bajaur reportedly having the highest immunization rates. The availability of vaccination cards improved in 2020 relative to 2012, and there was progress in the immunization status of children across all regions.

The quantitative findings highlighted the need to address misconceptions, raise awareness, and improve polio vaccine acceptance. This information can be used by policymakers and stakeholders to develop targeted strategies for preventing and controlling polio in the surveyed areas. By addressing knowledge gaps and ensuring access to immunization services, significant progress can be made towards eradicating polio in high-risk areas.

### 4.3. Qualitative Component

The focus group discussions and in-depth interviews yielded the qualitative component findings, which encompassed the perspectives of various respondents, including doctors, grandmothers, polio health workers, and media sources such as mosque announcements, TV, radio, IEC material, banners, and posters. The study participants highlighted a culture in which mothers are offered support from grandmothers in child-rearing, child health, and other pertaining matters.

“Grandmothers guide our daughters-in-laws about child-rearing and caring, counsel what to feed if the child has fever or pain, what type is food is good in hot and cold weather to keep the child healthy.”

Overall, the child’s health was regarded as not good by the study participants based on the compromised environmental conditions, socioeconomic status, negligence from parents and household members, poverty, and inflation.

“Health of our children is not good, as most children don’t get a vaccination, they don’t get proper nutrition, and live in unhygienic conditions. Nowadays, the diet of mothers is also not good that’s why children are weak.”

Subsequently, the study participants appreciated the polio workers for their efforts in creating awareness regarding child health and vaccination in the community.

“Polio workers give knowledge and guide us about health and vaccine, we are thankful to them that they visit our households repeatedly for the sake of community health.”

When specifically talking about routine immunization, the received responses were of a mixed nature. Most parents were aware of the importance of the polio vaccine and immunization for their children. However, despite several international- and national-level efforts, there exist negative health beliefs and vaccine hesitancy within the community. The respondents highlighted the perception of the adverse effect of vaccines on reproductive health as a prevalent misconception in the community.

“People believe that these vaccines are for family planning, cause infertility to decrease their generation.”

The findings also revealed that most respondents were knowledgeable about polio disease, but some wrongly perceived polio-related signs and symptoms. For example, it was shared that if children suffer from polio disease, they become physically weak, their hands and feet get twisted, and children become paralyzed.

“Only legs are not affected by polio disease but it may cause many other problems like weakness, low immunity, heart disease, indigestion, weakens of an arm or leg and it can also lead to death.”

The risk of contracting polio disease was believed to be increased if the child was weak, had poor health, lacked routine immunization against polio, and had frequent illnesses. One of the participants expressed how vaccine hesitancy led to the disrupted life of her niece.

“My niece got affected by polio disease because her mother never gave her polio drops. When she took her to the doctor, he told them that she got infected by poliovirus. Now, she is 18 years old but disabled. After this incident, my sister gave polio drops to every child to prevent them from polio virus.”

However, the polio vaccine and a clean and healthy environment were mentioned as the only preventive and safety measures.

“Water testing from drainage was done to identify the polio virus in our area and it has been proven that this is one of the main reasons for spreading this disease. Unfortunately, no vaccine and medicine can save children, until and unless the environment can be cleaned.”

The majority of respondents were aware of the OPV. Members of the community highly admired the rigorous efforts of polio workers as they addressed their queries and concerns and counseled and motivated them to vaccinate their children against polio. However, a misconception was found about its treatment and prevention. A few of the respondents were confused about whether polio is treatable. Some of them believed that every treatment is possible if treated in a timely manner.

“I have seen that there was one girl who was 18 months old, and she got polio attack and she became disabled. So, her family took her to the Syed family (the direct descendants of the Prophet Muhammad), and within 5 days she became normal. Another child was in the last stage, and they took the child to the Maulvi and then the child became normal.”

Shared refusal reasons for the polio vaccine were that some people are still against vaccination due to their misconceptions, doubts, fears, religious beliefs, family or cultural norms, limitations, preferences, fake news, and rumors.

“Mostly people refuse to give polio drops to their male child. When asked about the reason for refusal during the polio campaign, and said we have an only male child in our family and these drops cause impotence.”

In summary, the qualitative findings of the study provided insights from various stakeholders, including doctors, grandmothers, polio health workers, and media sources. Key findings highlighted the role of grandmothers in child-rearing, appreciation for polio workers’ hard work, concerns about poor child health, misconceptions about vaccine effects on fertility, and vaccine refusal based on religious beliefs and cultural norms. However, effective communication and education campaigns are needed to address misconceptions, build trust, and promote vaccine acceptance within the community.

## 5. Discussion

The study illustrated that while most parents had some knowledge about routine immunization and polio disease, the depth of understanding was limited. Many of the respondents did not have sufficient information regarding immunization and polio disease and lacked basic knowledge about polio-related signs and symptoms. These findings corroborated those of other studies conducted on the same subject [18,19]. Some of the respondents wrongly perceived polio as treatable. The common perceptions about polio transmission were that it occurs due to living in unhygienic conditions, not vaccinating children, and migrants with unknown polio vaccine status. Quite unlike misperceptions about the role of clergy or mosques in polio vaccination, sources of information shared by respondents included doctors, grandmothers, community health workers (CHWs)/lady health workers (LHWs), television, radio, information, education and counselling (IEC) material, banners, and posters.

Despite some negative perceptions, the overall opinion of respondents about vaccines was positive. Data evidence for routine immunization showed that most children under 5 years of age receive timely vaccinations for preventing diseases, except for refusals. Despite the rigorous collective efforts of the polio health workers and other stakeholders, the views among those refusing vaccines are still very rigid. However, reasons for refusal included wild conspiracy ideas about a hidden agenda to destroy the nation and the vaccine inducing infertility or not being halal [20]. These perceptions by a few also influence other community decision-makers and are the most critical barrier to overcome moving forward [21,22]. The corollary to the strength of the current anti-vaccination movements worldwide could not be starker.

There were differences in the way elders perceived health and immunization issues. Most of the grandmothers in the current survey considered their grandchildren as healthy compared to community elders who considered child health in their communities to be suboptimal. To our surprise, most participants could not relate childhood vaccination to optimizing a child’s health. There is much need to improve community understanding about the importance of childhood immunizations.

A substantial proportion of respondents displayed a negative attitude towards polio immunization. Community mobilization is a cornerstone of vaccine confidence and acceptance [18,23]. Analysis revealed that these mobilization activities are very helpful as initially people were rigid and did not have any understanding of vaccination, and even used to refuse to open their doors, but a drastic change was observed after a rigorous and collective effort by all stakeholders. The involvement of CHWs/LHWs positively impacted the vaccine coverage rate [24]. The steps of the CHWs and LHWs were highly admired as they were prominent in terms of communication and had good rapports with families through their day-to-day interactions with mothers. They also guided them in health and vaccine-related issues by addressing their concerns in a timely manner, which helped minimize their doubts and fears concerning vaccination. It was quoted that the number of zero doses has drastically declined. Studies suggest that developing a trusted relationship with parents is the key to influencing parental decision-making about polio and routine immunization [25].

Comparative data from surveys in 2012 and 2020 identified key gaps in the progress of the polio program in Pakistan. Although we found significant improvement in some areas, some areas need urgent attention. For example, the data on the perceived safety of the OPV is still concerning, as 25% of respondents in Karachi, 37% in Bajaur, and 48% in Pishin either considered the OPV as unsafe or were unaware of the safety of the OPV. This lack of knowledge was consequently reflected in OPV refusal rates, and we found a significant increase in refusals in Karachi and Bajaur. Furthermore, the comparative data also identified that a substantial proportion of children, 10% in Karachi, 5% in Bajaur, and 13.4% in Pishin, are unimmunized (zero doses), thus reflecting the compromised quality of routine immunization in these areas.

Religious and social beliefs are the most prominent barriers that prevent the disease from tipping over into complete eradication. We believe there is an utmost need for a coalition campaign involving religious scholars, civil society stakeholders, and media that could turn the tide against polio disease in Pakistan. Inadequate knowledge, false religious beliefs, and misconceptions are essential drivers of people’s understanding of polio disease. These factors are more likely to be the source of potential barriers to behavioral change that could help reduce the burden of polio in high-risk countries. Furthermore, there is a need to evaluate strategy effectiveness in the relationship between parents and health care providers, which is supported by online literature [18,26,27]. Additionally, polio-prevalent areas should be targeted to alter the negative attitudes of those who refuse vaccination through identification and subsequent counseling by community leaders/influential members, CHWs/LHWs, and other civil society stakeholders. Improving the attitude of the community along with their knowledge of polio will aid in turning the dream of a polio-free world into a reality for all children.

Our data on vaccine hesitancy and refusal can be explained by the fact that families mistrust and doubt the polio program, resulting in the recent upsurge in infection [2]. Most of these families are impoverished, underserved, and lack basic needs. Additionally, in recent years, social media has also played an important role in rapidly spreading false rumors about the polio program [28]. Furthermore, weakened essential immunization services, poor water and sanitation, and lack of maternal and child health care further compound the polio eradication challenge in Pakistan [29]. To overcome these challenges an integrated approach is required in which all of these services should be given to communities holistically. A large population-based trial conducted in high-risk areas for polio in Pakistan provided a community mobilization strategy. It targeted community-based health and immunization camps during polio immunization campaigns, which resulted in an effective increase in vaccine coverage in campaigns and routine immunization [30]. This model can be scaled up along along with integrated services to achieve maximum coverage for the OPV in high-risk areas.

Our study’s strength lies in its focus on the mixed-method approach, which sets the groundwork for subject areas for which the availability of literature is limited. Furthermore, the qualitative approach shed light on the respondents’ desirable responses, which enhanced data triangulation. Another key strength is the inclusion of participants from high-risk areas for polio. Additionally, the study results will assist stakeholders and health officials from the government and civil sectors in evaluating and modifying the current program’s effectiveness and policies regarding polio eradication in Pakistan. However, caution should be exercised as the study findings may not be generalizable countrywide. Participants were recruited using convenient purposive sampling, which may not account for variations within the population. Additionally, due to the study’s cross-sectional survey design, the establishment of a temporal relationship was not possible. To address potential barriers to behavioral change and reduce the burden of polio in high-risk countries, it is recommended that comprehensive education campaigns targeting parents and healthcare providers be implemented in both phases (i.e., planning and implementation phases). These campaigns should also emphasize the importance of vaccination, address misconceptions, and involve parents and healthcare providers. The strategy should employ various communication channels and platforms, such as social media, community outreach programs, and informational materials, in order to reach a wide audience. Furthermore, it should be culturally and contextually tailored to the specific cultural and socioeconomic contexts of countries at high risk of polio in order to maximize the impact. To evaluate the effectiveness of these strategies in enhancing the relationship between parents and healthcare providers, a comprehensive assessment should be carried out. By following these recommendations and continuously evaluating the strategy, decision-makers can develop an evidence-based approach to effectively reduce polio in high-risk countries.

## Figures and Tables

**Figure 1 vaccines-11-01206-f001:**
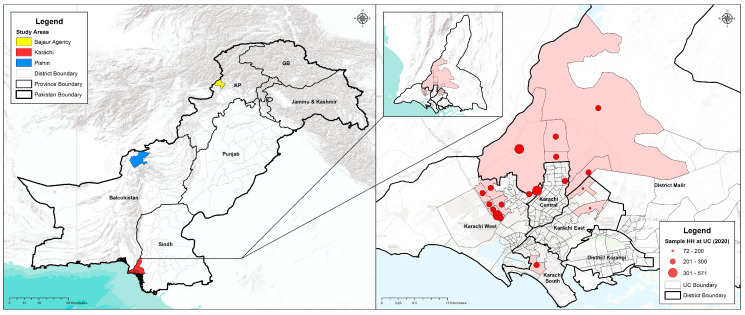
Survey areas in Pakistan and GIS locations of study areas in Karachi.

**Table 1 vaccines-11-01206-t001:** Study Sites and Sample Distribution of the Survey.

Karachi (18 UCs~210 HH/UC)	Bajaur (6 UCs~210 HH/UC)	Pishin (10 UCs~210 HH/UC)
Gujro, Songal, Maymarabad, Yousaf Goth, Mangopir, Godhra, Gulshan-e-Ghazi, Itehad Town, Islam Nagar, Nai Abbadi, Saeedabad, Muslim Mujahid Colony, Muhajir Camp, Islamia Colony, Pehlwan Goth, Metrovile Colony, Hijrat Colony, Baloch Goth	Khar, Mamund, Salarzai, Nawagai, Uttamkhel, Barang	Bazar Kona, Muchan, Bazar Pishin, Karbala, Batazai, Khanozai, Bostan, Dilsora, Malikyar, Kaza Villa

**Table 2 vaccines-11-01206-t002:** Distribution of Qualitative Components.

	Karachi		Bajaur		Pishin	
	FGDs	IDIs	FGDs	IDIs	FGDs	IDIs
Mothers	4	-	2	-	2	-
Fathers	4	-	2	-	2	-
Mothers-in-Law	4	-	2	-	2	-
Community influencers	-	8	-	4	-	4
Healthcare Providers	-	8	-	4	-	4
	12	16	6	8	6	8

**Table 3 vaccines-11-01206-t003:** Basic Demographic Information of Study Sites.

	Karachi	Bajaur	Pishin
Indicators			
Union councils	18	6	10
Households surveyed	3780	1258	2099
HH density	7.8	9.1	12.5
Total population	29,381	11,507	26,223
Male (%)	50.8	51	49.8
Total children under 5 years of age	6181	2144	4774
Male (%)	51.73	53.26	49.79
Illiteracy rate (among respondents)	44.1	74.7	84.2
Ownership of household (owned households)	61.3	92.6	81.6
Hand-washing practice (after defecation)	99.7	96.5	92.3
Water treatment (% treats water)	45.3	31.2	42.3

**Table 4 vaccines-11-01206-t004:** Knowledge and Perceptions of Polio and Information Sources.

	Karachi	Bajaur	Pishin
Indicators	(*n* = 3766)	(*n* = 1258)	(*n* = 2099)
Knowledge about Polio	*n* (%)	*n* (%)	*n* (%)
Yes	3748 (99.5)	1229 (97.7)	2036 (97)
No	18 (0.5)	29 (2.3)	63 (3)
Source of knowledge about Polio			
TV	2199 (47.1)	199 (15.8)	822 (39.2)
Radio	23 (0.5)	905 (71.9)	561 (26.7)
Newspaper	49 (1.1)	31 (2.5)	139 (6.6)
Posters	388 (8.3)	49 (3.9)	416 (19.8)
Masjid Imam	10 (0.2)	64 (5.1)	274 (13.1)
Local leaders	24 (0.5)	79 (6.3)	19 (0.9)
Elders	1231 (26.4)	330 (26.2)	526 (25.1)
Neighbors	368 (7.9)	169 (13.4)	445 (21.2)
Friends	131 (2.8)	66 (5.3)	264 (12.6)
Relatives	682 (14.6)	235 (18.7)	348 (16.6)
Doctor	592 (12.7)	629 (50)	451 (21.5)
Hakeem	0 (0)	1 (0.1)	29 (1.4)
Homeopathic doctor	0 (0)	0 (0)	14 (0.7)
Quack	0 (0)	0 (0)	16 (0.8)
LHV	13 (0.3)	28 (2.2)	26 (1.2)
Nurse	0 (0)	2 (0.2)	27 (1.3)
Traditional healer	0 (0)	0 (0)	8 (0.4)
LHWs	58 (1.2)	339 (27)	822 (39.2)
Vaccinators	36 (0.8)	364 (28.9)	83 (4)
Others	37 (0.8)	8 (0.6)	29 (1.4)
Consider Polio as a health problem			
Yes	3482 (92.5)	1088 (86.5)	1267 (60.4)
No	90 (2.4)	69 (5.5)	231 (11)
Don’t know	194 (5.2)	101 (8)	601 (28.6)
Knowledge about preventing a child from getting polio			
By proper disposal of waste/sewage	487 (10.4)	245 (19.5)	607 (28.9)
By vaccinating a child/person or polio drops	2643 (56.6)	811 (64.5)	1066 (50.8)
Avoiding contact with an infected child/person	86 (1.8)	96 (7.6)	305 (14.5)
Washing hands with soap and water	204 (4.4)	69 (5.5)	196 (9.3)
Others	997 (21.4)	6 (0.5)	9 (0.4)
Don’t know	426 (9.1)	139 (11.1)	438 (20.9)
Knowledge about how a child gets polio disease			
By drinking dirty water	602 (12.9)	246 (19.6)	701 (33.4)
By eating dirty food	624 (13.4)	318 (25.3)	543 (25.9)
Through sewage contamination of food/water	372 (8)	300 (23.9)	441 (21)
By air/breathing	34 (0.7)	117 (9.3)	452 (21.5)
By evil eye	48 (1)	212 (16.9)	442 (21.1)
Don’t know	1626 (34.8)	73 (5.8)	367 (17.5)
Allah’s will	793 (17)	452 (35.9)	842 (40.1)
Other	403 (8.6)	4 (0.3)	3 (0.1)
Knowledge about polio transmission			
By air/breathing	262 (5.6)	596 (47.4)	855 (40.7)
By evil eye	28 (0.6)	152 (12.1)	359 (17.1)
By contaminated water	389 (8.3)	183 (14.6)	496 (23.6)
By sewage contamination	707 (15.2)	191 (15.2)	697 (33.2)
Through polio vaccines	43 (0.9)	258 (20.5)	160 (7.6)
Don’t know	2543 (54.5)	334 (26.6)	773 (36.8)
Other	228 (4.9)	0 (0)	8 (0.4)
Knowledge about preventive measures for polio			
Completing routine EPI immunization	2485 (53.3)	900 (71.5)	1365 (65)
Completing routine EPI immunization plus getting your child vaccinated in NIDs	1327 (28.4)	248 (19.7)	473 (22.5)
Proper nutrition of the child	847 (18.2)	116 (9.2)	561 (26.7)
Proper hygiene	1331 (28.5)	288 (22.9)	226 (10.8)
Proper care	510 (10.9)	138 (11)	138 (6.6)
Keeping child away from a sick child	147 (3.2)	302 (24)	651 (31)
Proper sewage disposal and sanitation	99 (2.1)	75 (6)	178 (8.5)
Other	429 (9.2)	0 (0)	7 (0.3)
Do not know	282 (6)	57 (4.5)	247 (11.8)

**Table 5 vaccines-11-01206-t005:** Knowledge & Perceptions of Polio Vaccines.

	Karachi	Bajaur	Pishin
Indicators	(*n* = 3766)	(*n* = 1258)	(*n* = 2099)
Knowledge about oral polio vaccine	*n* (%)	*n* (%)	*n* (%)
Yes	3759 (99.8)	1253 (99.6)	2044 (97.4)
No	7 (0.2)	5 (0.4)	55 (2.6)
Knowledge about OPV protecting against polio			
Yes	3347 (88.9)	1186 (94.3)	1445 (68.8)
No	55 (1.5)	21 (1.7)	25 (1.2)
Don’t know	71 (1.9)	30 (2.4)	342 (16.3)
Not sure	293 (7.8)	21 (1.7)	287 (13.7)
Knowledge about the consequences of not giving OPV			
Child gets polio	2419 (64.2)	669 (53.2)	1208 (57.6)
Child will not get polio	919 (24.4)	359 (28.5)	492 (23.4)
Polio would not be eradicated from Pakistan	172 (4.6)	111 (8.8)	191 (9.1)
Don’t know	256 (6.8)	119 (9.5)	208 (9.9)
Perceptions about the safety of OPV			
Completely safe	2891 (76.8)	791 (62.9)	880 (41.9)
Reasonably safe	517 (13.7)	353 (28.1)	405 (19.3)
Not safe at all	179 (4.8)	16 (1.3)	199 (9.5)
Other	4 (0.1)	0 (0)	0 (0)
Don’t Know	175 (4.7)	97 (7.7)	615 (29.3)
Ever refused to give polio drops to your child?			
Never	2897 (76.9)	943 (75)	1689 (80.5)
Once	294 (7.8)	209 (16.6)	111 (5.3)
More than once	420 (11.2)	92 (7.3)	290 (13.8)
Always	137 (3.6)	14 (1.1)	5 (0.2)
Other	18 (0.5)	0 (0)	4 (0.2)
Are all your children immunized with OPV?			
Yes	3543 (94.1)	1162 (92.4)	1996 (95.1)
No	95 (2.5)	16 (1.3)	33 (1.6)
Partially	99 (2.6)	32 (2.5)	42 (2)
Not at all	20 (0.5)	3 (0.2)	2 (0.1)
Don’t know	9 (0.2)	45 (3.6)	26 (1.2)
Did your child/children receive OPV during the recent campaign			
Yes	3526 (93.6)	1241 (98.7)	2067 (98.5)
No	238 (6.3)	14 (1.1)	18 (0.9)
Don’t know	2 (0.1)	3 (0.2)	14 (0.7)
Decision-maker regarding the health matters of the children?			
Mother-in-law	419 (9)	353 (28.1)	769 (36.6)
Father-in-law	136 (2.9)	359 (28.5)	633 (30.2)
Husband	2304 (49.4)	763 (60.7)	1812 (86.3)
Mother/Myself	2212 (47.4)	259 (20.6)	363 (17.3)
Other	65 (1.4)	0 (0)	1 (0.1)
Decision-maker regarding immunization of the children?			
Mother-in-law	336 (8.9)	158 (12.6)	201 (9.6)
Father-in-law	92 (2.4)	256 (20.4)	244 (11.6)
Husband	1562 (41.5)	703 (55.9)	1593 (75.9)
Mother/Myself	1717 (45.6)	139 (11.1)	59 (2.8)
Others	59 (1.6)	2 (0.2)	1 (0.1)

**Table 6 vaccines-11-01206-t006:** Reasons for Refusal of OPV (Ever Refused).

	Karachi	Bajaur	Pishin
Reasons			
Reasons for refusing polio drops to child?	*n* (%)	*n* (%)	*n* (%)
Vaccine is not halal	21 (2.4)	10 (3.2)	187 (45.6)
Vaccine can cause sterility	163 (18.8)	62 (19.7)	280 (68.3)
Vaccine is not safe	162 (18.6)	11 (3.5)	161 (39.3)
Child has received polio drops too many times	31 (3.6)	67 (21.3)	93 (22.7)
Prohibited by community/local leaders	0 (0)	0 (0)	4 (1)
Against my religious belief	6 (0.7)	0 (0)	19 (4.6)
Other	227 (26.1)	42 (13.3)	13 (3.2)
Total	869	315	410
Reasons for not giving OPV in the last polio campaign			
Polio team did not visit the house	86 (26.5)	90 (86.5)	92 (83.6)
Did not know about OPV	53 (16.3)	6 (5.8)	5 (4.6)
Went, but the distribution point had run out of OPV	3 (0.9)	1 (1)	4 (3.6)
Child was not at home at the time of the visit	2 (0.6)	0 (0)	1 (0.9)
Child was sick	13 (4)	2 (1.9)	1 (0.9)
Distribution site is far away	24 (7.4)	0 (0)	1 (0.9)
Family members/elders did not allow the child to get OPV	99 (30.5)	5 (4.8)	5 (4.6)
Other	45 (13.9)	0 (0)	1 (0.9)
Don’t Know	86 (26.5)	90 (86.5)	92 (83.6)
Total	223	96	103

**Table 7 vaccines-11-01206-t007:** Comparison of KAP Surveys in 2020 and 2012.

	Karachi		Bajaur		Pishin	
Indicators	2012	2020	2012	2020	2012	2020
Respondent’s knowledge about polio	94.1	99.5 *	97.5	97.7	98.4	97.0
Respondent considers polio to be a health problem	92.2	92.5	85.6	86.5	77.6	60.4 *
Respondents having knowledge about OPV	99.3	99.8	96.8	99.6	97.8	97.4
Perceptions about the safety of OPV						
Completely safe	76.5	76.8	61.6	62.9	49.7	41.9
Reasonably safe	11.2	13.7	11	28.1	28.9	19.3
Not safe at all	1.1	4.8	2.7	1.3	1.4	9.5
Don’t Know	11.2	4.7	24.7	7.7	19.7	29.3
Ever refused OPV	5.0	23.1 *	17.4	25.0 *	20.0	19.5
Immunization card available	26.4	47.7	12.9	78.1	6.9	44.5
Immunization status						
Fully immunized	52.3	55.5 *	28.3	67.8 *	56.5	68.6 *
Partially immunized	27.9	35.1	35.4	27.3	17.2	18.0
Unimmunized	19.8	9.4 *	35.1	5.0 *	25.6 *	13.4

* Represents significant difference *p* < 0.05.

## Data Availability

The dataset of the study is available upon request from the corresponding author.

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
