# Peer review of "Exploring Knowledge and Perceptions of Polio Disease and Its Immunization in Polio High-Risk Areas of Pakistan"

_vaccines, 2023, doi:10.3390/vaccines11071206_

Round 1
Reviewer 1 Report
Interesting research results, but also forecasting a number difficulties. The authors have done an excellent job and definitely should continue it. I am curious if they see any solutions to improve the vacccination program against polio. For publication.
Author Response
Dear Reviewer,
Thank you for your invaluable contribution to improving our research.
We have thoroughly revised the manuscript based on the valuable feedback. Attached is the updated version for your reference. We greatly appreciate your time and expertise, and we eagerly await your feedback on the revised manuscript.
Regards,
Sajid Soofi

Reviewer 2 Report
Dear Author,
1 For example, misperceptions about the oral polio vaccine repre-53 sented by religious leaders and extremist groups and organizations in Pakistan have sug-54 gested major inhibiting factors for polio vaccine uptake [12, 13, and 14]. In addition, ru-55 mors such as OPV being anti-Islamic, having haram impermissible ingredients, being a 56 cause of infertility and the GPEI program being a foreign agenda are prevailing in the 57 high-risk groups [11-14].(Please revise these lines
2For the qualitative component, we conducted FGDs and IDIs in Karachi, Pishin, and 99 Bajaur. Respondents for the qualitative component included; decision-makers at the 100 household level (including fathers, mothers, and mothers in law), Influencers at the com-101 munity level prominent members of the community and health care providers, including 102 the community health workers and vaccinators (Web Table 2)(Correct these with updated literature
3 Mixed Method Study to Assess the Knowledge and 2 Perceptions around Polio and Its Immunizationshould be highlighted)
4For data collection, we developed a structured questionnaire for the household sur-89 vey. The survey instrument was translated into the national language, Urdu and back-90 translated into English. The survey instrument was pre-tested in areas not part of the sur-91 vey sample and updated as per the findings during the pre-test. We hired local teams of 92 female data collectors who were accustomed to the local culture and language and had 93 prior data collection experience. We conducted a four-day centralized training for the data 94 collectors on survey methodology, sampling technique, and survey instrument followed 95
by two days of mock field interviews. After the training, the survey teams collected data 96 at the household level. Written consent was taken from the respondents (mother, father, 97 or the caregiver) before all interviews. 98For the qualitative component, we conducted FGDs and IDIs in Karachi, Pishin, and 99 Bajaur. Respondents for the qualitative component included; decision-makers at the 100 household level (including fathers, mothers, and mothers in law), Influencers at the com-101 munity level prominent members of the community and health care providers, including 102 the community health workers and vaccinators (Web Table 2). We used the purposive con-103 venience sampling technique for selecting participants for in-depth interviews (IDIs) and 104 focus group discussions (FGDs). Similar to the quantitative component we developed and 105 pre-tested data collection tools (the FGD and IDI guidelines) by conducting FGDs and 106 IDIs with various target groups to assess the clarity and relevance of the content. The qual-107 itative data was collected by an expert team comprised of a female moderator and note-108 taker supported by the local facilitators. (Please revise the above paragraph)
5. T a b l e nos are not correct and not in a order please arrage as per journal format
6 Please arrange references by using zotero or mendeley as per journal guide lines
7 Plaggiarism should be checked
NIL
Author Response
Dear Reviewer,
We have thoroughly revised the manuscript based on your valuable feedback by addressing all the provided comments. Attached is the updated version for your reference. We greatly appreciate your time and expertise, and we eagerly await your feedback on the revised manuscript.
Thank you for your invaluable contribution to improving our research.
Regards,
Sajid Soofi

Reviewer 3 Report
Comments are included in the manuscript-pdf

Author Response

(The authors gave the same response as above.)

Round 2
Reviewer 2 Report
Dear authors,
please check tables and information please revise standard format of content in tables and please include conclusion
references still minor revision is required
tion.org/who-we-are/our-mission/. 550
2. Programme PPE. Polio in Pakistan [Available from: https://www.endpolio.com.pk/polioin-pakistan. 551
3. Organization WH. Pakistan: Polio Eradication Initiative [Available from: https://www.emro.who.int/pak/pro-552 grammes/polio-eradication-initiative.html. 553
4. Initiative GPE. Where We work [Available from: https://polioeradication.org/where-we-work/. 554
5. Khowaja AR, Zaman U, Feroze A, Rizvi A, Zaidi AK. Routine EPI coverage: subdistrict inequalities and rea-555 sons for immunization failure in a rural setting in Pakistan. Asia pacific journal of public health. 2015;27(2):NP1050-556 NP9. 557
6. National EPI Policy and Strategic Guidelines -Pakistan. Ministry of National Health Services; 2015. 558
7. Organization WH. Global emergency action plan 2012–2013. 2012. 559
8. Kew O. Reaching the last one per cent: progress and challenges in global polio eradication. Current opinion in 560 virology. 2012;2(2):188-98. 561
9. Obregón R, Chitnis K, Morry C, Feek W, Bates J, Galway M, et al. Achieving polio eradication: a review of 562 health communication evidence and lessons learned in India and Pakistan. Bulletin of the World Health Organization
Best regards
its fine minor revision in english language is needed in introduction and methodology
and conclusion .ENGLUISH LANGUAGE some corrections
Author Response
Thank you for your valuable feedback. We have made the necessary changes. Please find the revised file appended for your reference.
Regards,
Sajid Soofi
